# Fire Hazard Analysis on Different Fire Source Locations in Multi-Segment Converging Tunnel with Structural Beams

**Lixin Wei** [1], **Honghui Tang** [1], **Jiaming Zhao** [2], **Shiyi Chen** [3], **Yiqiang Xie** [1], **Shilin Feng** [3], **Zhisheng Xu** [3] and **Zihan Yu** [3,*]

1  Guangzhou Municipal Engineering Design & Research Institute Guangzhou Co., Ltd., Guangzhou 510000, China; weilx@gzmedri.com (L.W.); tanghh@gzmedri.com (H.T.); xieyq@gzmedri.com (Y.X.)
2  CABR Fire Safety Science & Techology Co., Ltd., Beijing 100013, China; zhaojiaming93@126.com
3  School of Civil Engineering, Central South University, Changsha 410075, China; 224811145@csu.edu.cn (S.C.); 234812296@csu.edu.cn (S.F.); zhshxu@csu.edu.cn (Z.X.)
*  Correspondence: 224801064@csu.edu.cn

**Abstract:** To investigate the fire risk in a complex tunnel with varying cross-sections, sloped structures, and dense upper cover beams, this study considered four fire source positions: the immersed tube section, confluence section, highway auxiliary road section, and four-lane sections of the main line. It also considered four beam spacings: 1 m, 1.8 m, 3.6 m, and 7.2 m. The Fire Dynamics Simulation Software FDS was utilized to create a comprehensive tunnel model. The analysis focused on temperature and visibility changes at a 2 m height under a 20 MW fire condition for different fire source positions. These changes were then compared with critical danger values to assess the safety of evacuating personnel within the tunnel. Subsequently, this study proposed corresponding emergency rescue strategies. The findings indicated that when the beam grid spacing exceeded 3.6 m, the upper dense beam gap showed a robust smoke storage capacity, leading to a reduced distance of high-temperature smoke spread. However, this increased smoke storage disrupted the stability of the smoke layer, resulting in a heightened smoke thickness. The location of the ventilation vent at the entrance of the immersed tunnel section caused a non-uniform ventilation flow under the girder, deflecting the smoke front towards the unventilated side and decreasing visibility in the road auxiliary area. In comparison to scenarios without a beam lattice, the presence of a beam lattice in the tunnel amplified fire hazards. When the beam lattice spacing was 3.6 m or greater, the extent of the hazardous environment, which is unfavorable for personnel evacuation, expanded. With the exception of the scenario where the fire source was located in the highway auxiliary roadway, all other conditions surpassed 150 m, which is roughly one-third of the tunnel length. Consequently, more targeted strategies are necessary for effective evacuation and rescue efforts.

**Keywords:** tunnel fire; structural beams; longitudinal temperature distribution; visibility

## 1. Introduction

In recent years, China's rapidly developing economy has led to a growing transport industry and increased traffic, resulting in various safety issues. The government actively promotes road tunnel construction to support sustainable transportation development and address land resource constraints. With the expanding network of road tunnels, tunnel accidents are rising. Among these, tunnel fires are the most common and have severe consequences. The confined structure of tunnels makes it challenging to effectively disperse smoke and fire byproducts, necessitating longitudinal ventilation systems.

In the study of tunnel fire characteristics, the initial analysis must focus on the fire source. The fire source's location is a pivotal factor that significantly influences tunnel fires. Scholars have investigated the impact of different lateral and longitudinal positions of fire sources, as well as the fire source height, on fire parameters, such as the high-temperature

smoke spread and temperature distribution in tunnel fires. Chen et al. [1] studied smoke characteristics during one-dimensional horizontal diffusion stages in tunnels with different fire locations. They discovered a direct relationship between the average smoke mass flow, the fire-to-sidewall distance, and the heat release rate (HRR). They also proposed a prediction model for the mean smoke mass flow rate during this stage. Zhong et al. [2] examined how various transverse fire positions affect smoke return suppression in tunnel fires through modeling experiments. They found that the mass burning rate of the fire source remained consistent, while the maximum temperature rise under the ceiling increased exponentially as the flame neared the sidewalls. Huang et al. [3] examined the impact of the fire source's longitudinal position on the maximum ceiling gas temperature. They noted an asymmetric ceiling gas temperature distribution and established an empirical equation to track the maximum ceiling temperatures based on the fire source's longitudinal position. Zhao et al. [4] used a small-scale tunnel model to investigate the natural ventilation in tunnel fires. They focused on the impact of the fire's position along the tunnel and pinpointed the most critical position, which is valuable for designing tunnel fire ventilation systems. Haddad et al. [5] examined how various horizontal fire positions affect temperature diffusion, particularly the stratification of the maximum smoke temperature under the ceiling. They observed a consistent trend in the typical temperature profile corresponding to the fire source's position. Gannouni [6] conducted CFD numerical simulations to investigate the variation in the critical wind speed in diverse fire scenarios with varying fire source heights above the ground. The results indicate an inverse relationship, with higher fire source heights corresponding to lower critical wind speeds and a resultant increase in the temperature rise of smoke below the ceiling. Relevant formulas for predicting critical wind speeds were established. Barbato et al. [7] conducted a comprehensive literature review on the fire safety of highway tunnels, emphasizing the crucial role of computational fluid dynamics in tunnel fire research. The survey results underscore the complexity of capturing all factors related to the critical wind speed in a single formula. Consequently, multiple critical wind speed prediction formulas have been developed, incorporating external factors, like the tunnel geometry and slope.

The gas temperature at the tunnel's ceiling is a critical indicator for assessing tunnel fire risks, with direct implications for people's safety and well-being inside the tunnel. Xu et al. [8] investigated the hot smoke layers, temperature distribution, and maximum temperatures in linear fire sources. They developed empirical formulas to predict temperature distributions in different-shaped linear fires. Guo et al. [9] studied the radiant heat flow and temperature distribution in tunnel fires affected by longitudinal wind and sidewall constraints. They proposed a thermal radiation model based on the solid flame assumption and confirmed its accuracy through experiments. Li et al. [10] experimented with the impact of a fire's longitudinal position on the hot soot temperature under the ceiling. They proposed an attenuation model considering the fire's longitudinal position in branch tunnels. Qiu et al. [11] examined the maximum ceiling temperature and transverse temperature distribution in situations involving dual transverse fire sources. They developed a novel model to better assess the fire risk in tunnels with these conditions. Liu et al. [12] experimentally studied maximum temperatures and ceiling temperature changes in naturally ventilated tunnels with transverse cross-channels. They found that fires at bifurcation points had higher maximum temperatures and introduced a model for predicting temperature changes in the cross-channels.

To enhance space utilization and accessibility, complex road tunnels have become more prevalent, including bifurcated tunnels, inclined tunnels, top openings, and adjacent tunnels. Both domestically and internationally, researchers have studied the fire characteristics of these complex tunnel designs. Various parameters were involved, such as the longitudinal temperature distribution, smoke spread distance, smoke layer thickness, smoke backflow length, longitudinal temperature profile, temperature decay coefficient, critical wind speed, heat flux, and so on. Ura et al. [13] studied the smoke diffusion in shallow urban road tunnels with top openings. Their modeling experiments revealed a

constant smoke diffusion distance under the experimental conditions, irrespective of the fire size. Notably, smoke along the ceiling rapidly dissipated when the maximum opening rate reached 15%. Wang et al. [14] studied top-opening tunnel fires using theoretical analysis and full-scale experiments. They developed a predictive model for the smoke return distance. The results from these experiments and analyses can inform tunnel fire research, offering a scientific foundation for fire prevention and the construction of top-opening road tunnels. Tanaka et al. [15] investigated smoke characteristics in urban shallow road tunnels with top openings during fires, noting that favorable lateral exterior wind conditions resulted in shorter smoke dispersion distances. These distances remained constant across a range of fire heat release rates. Chen et al. [16] investigated smoke characteristics in bifurcated tunnel fires. Their experiments revealed that the fire source's location impacts the ceiling temperature, while increased longitudinal ventilation rates result in a linear decrease in the maximum tunnel temperature. Lu et al. [17] used numerical simulations to examine the ceiling temperature distribution in curved tunnels during fires. They observed that temperature decay coefficients increased linearly with the tunnel curvature, with less dependence on fire heat release rates, and introduced a new exponential decay model for predicting longitudinal temperatures in various tunnel curvature situations. Chow et al. [18] examined the smoke movement in inclined tunnels during fires with longitudinal ventilation and proposed a modified empirical formula for the critical velocity based on experiments and calculations. Zhao et al. [19] conducted 1:16 scale model experiments to investigate the influence of external wind on the lateral smoke flow between adjacent tunnel fires. The results uncovered a robust correlation between the critical lateral flow speed, tunnel cross-section, separation distance between adjacent tunnels, and fire heat release rate, contributing to the formulation of relevant prediction formulas. Zhu [20] studied fire heat fluxes in horseshoe and circular tunnels with longitudinal ventilation at a critical wind speed. They developed a fast predictive model for heat fluxes based on numerical simulations and theoretical research.

With the rapid development of cities and the scarcity of land resources, a unique construction model has emerged where tunnels are integrated with above-ground structures, such as building green parks or large commercial buildings on the tunnel covers, as shown in Figure 1. In these cases, the strategic arrangement of beam and column structures in the tunnel ceiling ensures the integrity of the construction, supporting the weight of development projects above ground. This symbiotic relationship between underground transportation infrastructure and above-ground urban development showcases the versatility and ingenuity of contemporary engineering solutions.

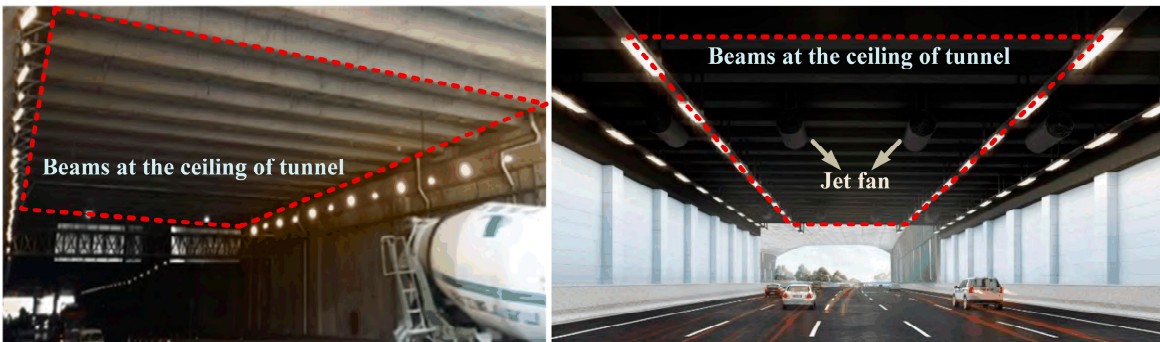

**Figure 1.** Example of a project with a superstructure at the top of a tunnel.

Currently, some scholars are focusing on the unique tunnel structure with beams at the top, conducting research on the characteristics and patterns of the smoke spread during fires in such structures. Tian et al. [21] used model experiments and numerical simulations to study the effect of top beams on the airflow and smoke in tunnels with virtual fire sources. They developed a critical velocity prediction model for top beam tunnels to aid in smoke control system design. Chaabat et al. [22] explored the influence of the tunnel

height (H) as a reference dimension on smoke spread characteristics in longitudinally ventilated tunnels, considering small beams (H/10) and large beams (H/4, H/3) during fire incidents. The findings suggest that larger beams prove more effective, successfully suppressing the smoke backflow even at exceptionally low critical wind speeds. Halawa and Safwat [23] employed FDS numerical simulations to examine the obstruction of the smoke spread by obstacles at the top of tunnels, considering beam heights of 0.5 m and 1 m, with beam spacings of 5 m, 10 m, 15 m, and 20 m. The results demonstrated that, compared to scenarios without beams, under conditions with a beam height of 1 m and a beam spacing of 5 m, the distance of the smoke spread was reduced by 66.5%. Zhang et al. [24] used FDS numerical simulations to explore the impact of inclined smoke barriers above an immersed tube tunnel on lateral smoke exhaust efficiency. Three barrier heights (1 m, 2 m, 3 m) were considered, and the results indicated that higher barrier heights led to an improved smoke exhaust performance in smaller-scale fire incidents.

This study adopts the *SFPE Fire Engineering Handbook* [25] guidelines, setting 10 m as the standard for defining the safety visibility range. Additionally, to ensure a secure evacuation and minimize the radiative heat effects of high-temperature smoke, a criterion of 60 °C at a 2 m height is established. While the existing literature on tunnel fires predominantly focuses on various tunnel configurations, there is a notable research gap concerning tunnels with covers. It is crucial to acknowledge that structures such as beams on top of covered tunnels significantly impact the longitudinal ventilation, smoke extraction, and temperature distribution during a fire. In order to provide data support for the smoke control design of complex tunnels with covered dense beams and to offer scientific theoretical guidance for the formulation of emergency evacuation and rescue strategies, we conducted a study on the fire hazards and temperature decay characteristics of such tunnels.

## 2. Materials and Methods

### 2.1. Model Settings

This paper employs the Fire Dynamic Simulator (FDS), a simulation program developed by the National Institute of Standards and Technology (NIST) in the USA, to conduct a numerical simulation study. Utilizing the Large Eddy Simulation (LES) method within the turbulence model, it effectively addresses the interaction between turbulence and buoyancy. This enables a more precise simulation of the flue gas flow pattern in a narrow channel driven by buoyancy.

The investigation of fire risks and temperature variation in complex tunnels with upper covers focused on various ignition source locations. The model was based on an actual tunnel project situated on the upper cover of the Pearl River in Guangzhou, China. The tunnel intersected with an auxiliary road through an underwater immersed tube section. On top of the tunnel, there was a convention and exhibition center spanning 328 m. A comprehensive numerical simulation was conducted using FDS 6.7.5 software, resulting in a complex model due to its integration with the actual project. The tunnel model was divided into four sections, each with specific dimensions, as shown in Figure 2. These sections included an immersed tube tunnel section measuring 100 m in length, 7.5 m in width, and 5 m in height, with a 4% gradient; the confluence section, which was 104 m long and 7.5 m wide, with heights varying from 9.2 m to 7.12 m; a highway auxiliary convergence area, also 104 m long and 6.5 m wide, with dimensions ranging from 3.7 m to 7.12 m; another section of the highway auxiliary convergence, which was 104 m long and 6.5 m wide, with heights between 3.7 m and 7.12 m; and a four-lane tunnel area, which was 224 m long, 15 m wide, and 7.12 m high. Except for the underwater immersed tube tunnel section, all other parts were located beneath the convention and exhibition center. In the tunnel area above, there were dense upper cover beams with cross beams spaced at 7.2 m and a width of 0.4 m. The longitudinal girder spacing was 9.6 m with a width of 0.4 m.

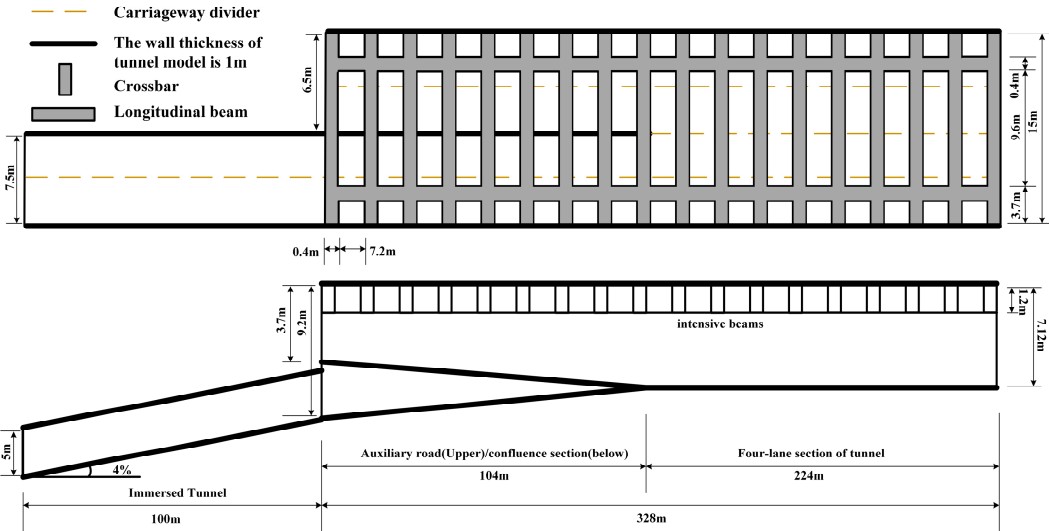

**Figure 2.** Schematic diagram of the dense beam (front and top views).

### 2.2. Boundary Conditions

Table 1 lists the detailed parameter settings for the numerical simulations used in this paper. On one side of the tunnel, there was a supply air intake surface at the entrance of the immersed tube tunnel, with a ventilation wind speed of 2 m/s. The other opening surfaces were completely open, with no wind speed, as depicted in Figure 3. Temperature and visibility measurement points were situated at a 2 m height within the tunnel. The spacing of these measurement points varied based on the fire source's location: 1 m spacing in the vicinity of the fire source, 2 m spacing in the section close to the fire source, and 5 m spacing in the area far from the fire source. Figure 4 illustrates the four locations of the fire source, with an example placement in the immersed tube section, visually depicting the measurement point locations. Figure 5 shows the positioning of the measurement points and the fire source. The top view and three temperature and flow velocity section locations were indicated to obtain temperature and flow velocity data in the longitudinal section at the highway auxiliary road's side and the middle of the immersed tube section's side. A transverse section at the tunnel entrance beneath the girder was also considered. The fire source was represented as a rapid $t^2$ fire with a 20 m$^2$ area, a heat release rate of 1000.0 kW/m$^2$ per unit area, and a total fire source power of 20 MW, as detailed in Figure 5. It is important to note that when the fire source was positioned in the four-lane section, the measurement point was centered within the tunnel.

**Table 1.** Numerical simulation parameter settings.

| Parametric | Settings |
|:---|:---:|
| Longitudinal ventilation velocity | 2 m/s |
| Heat release rate | 20 MW ($t^2$, 326 s) |
| Environmental temperature | 293 K |
| Gravitational acceleration | 9.8 m/s$^2$ |
| Surface materials | INERT |
| Simulation time | 1800 s |

### 2.3. Grid Division

To ensure the accuracy of the numerical simulations, the grid size was analyzed, theoretically identifying the largest-scale vortex structure within the fire plume. This approach balanced accuracy with computational efficiency. In this study, with a 20 MW fire source, a grid size ranging from 0.198 to 0.793 was determined. The division of grid regions is illustrated in Figure 6. Given the complexity of the model, it was segmented into

seven grid regions labeled as mesh 01 to mesh 07. The selection of the most precise mesh sizes was crucial to maintain model and calculation accuracy. Specifically, a mesh size of 0.2 m, as recommended, was chosen for the densely beamed lattice area on top. For the lower part of the tunnel, a slightly larger mesh size of 0.4 m was utilized. This resulted in a total of 2,542,250 mesh units.

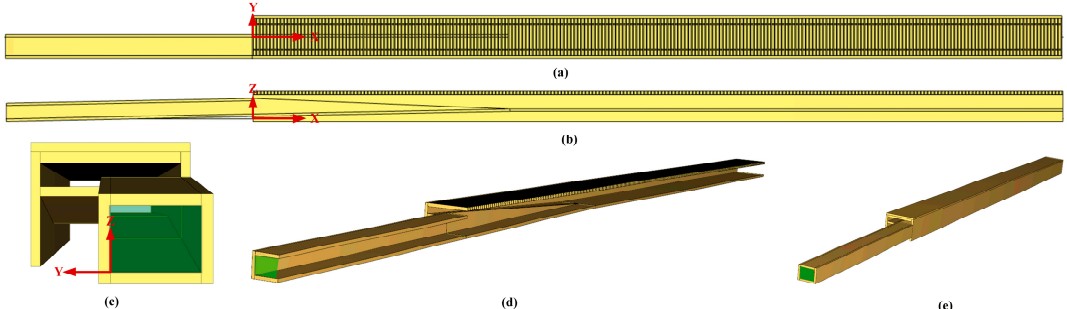

**Figure 3.** Schematic diagram of the model tunnel: (**a**) Schematic of modeled tunnel with dense beams (top view); (**b**) Schematic of the internal structure of the modeled tunnel (front view); (**c**) left view; (**d**) Schematic of the internal structure of the modeled tunnel; (**e**) Model Tunnel Overview.

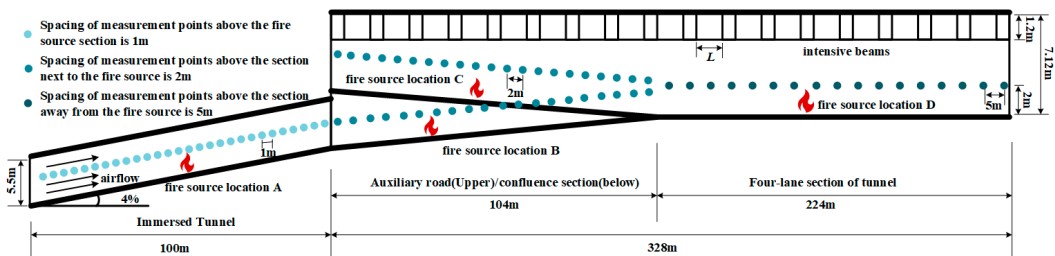

**Figure 4.** Schematic diagram of the tunnel model and measurement point arrangement (front view; dimensions are internal).

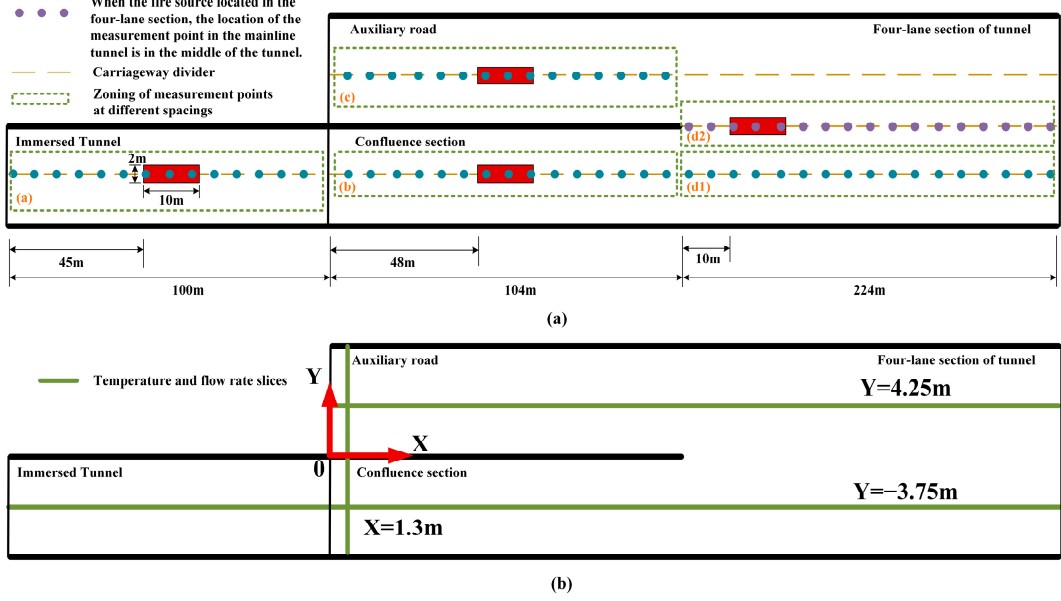

**Figure 5.** Schematic diagram of the location of temperature measurement points, ignition sources, and temperature flow rate slices (top view): (**a**) measurement points and ignition source locations; (**b**) Temperature and flow rate slicing schematic.

$$D^* = \left( \frac{\dot{Q}}{\rho_\infty c_p T_\infty \sqrt{g}} \right)^{2/5} \tag{1}$$

$$\frac{D^*}{\delta_x} \approx 4 \sim 10 \tag{2}$$

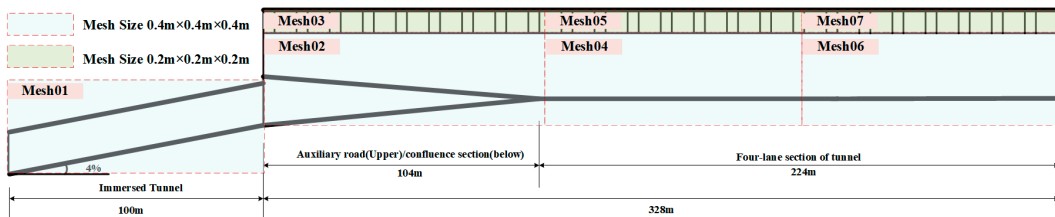

**Figure 6.** Schematic diagram of grid demarcation.

$D^*$ is the characteristic diam of the fire source, m; $\dot{Q}$ is the heat release rate of the fire source, kW; $\rho_\infty$ is the ambient air density, with a value of 1.2 kg/m$^3$; $c_p$ is the specific heat of air the at constant pressure, with a value of 1.014 kJ/kg·K; $T_\infty$ is the ambient temperature, 293 K; $g$ is the acceleration of gravity, 9.8 m/s$^2$; $\delta_x$ is the nominal size of the grid cell, m.

### 2.4. Working Condition Settings

The complex tunnel section structure was the focal point of this study. It examined four distinct fire source locations within the tunnel: the immersed tube section, the immersed tube confluence section, the highway auxiliary road section, and the four-lane tunnel section. To investigate the impact of the presence and spacing of dense beams on the tunnel temperature and visibility, four beam configurations were considered: no beams, 1 m spacing, 1.8 m spacing, 3.6 m spacing, and 7.2 m spacing. This resulted in a total of 20 simulations. The specific configuration of the working conditions is outlined in Table 2.

**Table 2.** Summary of working condition settings.

| No | Fire Location | HRR/MW | Velocity/m/s | Beam Spacing/m |
|----|---------------|--------|--------------|----------------|
| A0–A4 | Immersed tunnel | | | No beam, 1, 1.8, 3.6, 7.2 |
| B0–B4 | Confluence section | | | No beam, 1, 1.8, 3.6, 7.2 |
| C0–C4 | Auxiliary road | 20 | 2.5 | No beam, 1, 1.8, 3.6, 7.2 |
| D0–D4 | Four-lane section tunnel | | | No beam, 1, 1.8, 3.6, 7.2 |

## 3. Results and Discussion

### 3.1. Temperature Distribution in the Tunnel at Different Fire Locations

As shown in Figure 7a, when the fire source was located in the immersed tube section, the temperature directly above it rapidly rose to 770 °C. Within a 10 m radius of the fire, the temperatures decreased significantly. Upstream of the fire source, the temperature returned to ambient levels, indicating that a longitudinal air velocity of 2 m/s effectively prevented the backflow of flue gas.

In the confluence section, wider beam spacing resulted in higher longitudinal temperatures. This was due to the smoke originating from the buried area, causing turbulence in the region. Larger beam spacing intensified this turbulence, disrupting the settling of high-temperature smoke, resulting in higher temperatures compared to scenarios with smaller beam spacing. As the smoke progressed into the four-lane section, its temperature gradually decreased from 65 °C to the ambient temperature.

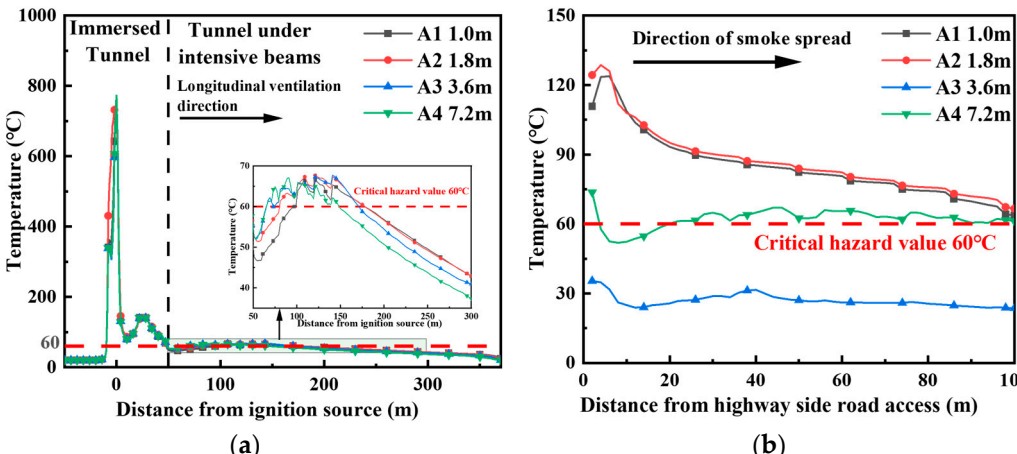

**Figure 7.** Longitudinal temperature distribution at 2 m height of tunnel (fire source located in the immersed tunnel): (**a**) tunnel mainline section, the green area shows the details of the change; (**b**) auxiliary road section.

In Figure 7b, it was observed that temperatures within 50 m of the highway aisle entrance were higher compared to the confluence section. This was due to the smaller spacing between the highway aisle and the top plate. Even with the same thickness of the smoke layer, the 2 m height in the highway aisle experienced higher temperatures. The highway aisle was naturally ventilated, so the smoke temperature was not affected by the cooling effect of ventilation.

The effect of beam spacing on the longitudinal temperature distribution at a 2 m height in the highway aisle and the confluence section exhibited an opposite pattern, where smaller beam spacing resulted in higher temperatures at the highway aisle location. In essence, the results highlighted the significant impact of beam spacing on temperature distribution and how different configurations influenced temperature profiles within the tunnel.

As shown in the temperature distribution cloud diagram in Figure 8, it was observed that upstream of the fire source there was no smoke backflow, and temperatures remained close to ambient. Moving downstream from the fire source, temperatures gradually decreased with the increasing distance. In the immersed tube tunnel, which featured a 4% slope, and at the confluence section interface, with a sudden cross-section change, complex airflow conditions resulted due to boundary effects. Factors like the chimney effect and longitudinal wind speed accelerated the smoke spread, forming a low-temperature vortex region, as depicted in Figure 8c.

As the smoke progressed into the confluence section and the highway auxiliary road section, thermal buoyancy caused it to rise, hit the tunnel ceiling, and spread laterally to the highway auxiliary road while also moving longitudinally along the tunnel ceiling. Increasing the spacing between the beams enhanced the smoke storage capacity and shortened the distance over which high-temperature smoke spreads, providing better control. This study delved into the dispersion of smoke and the distribution of temperature at a 2 m height, considering a beam height of 1.2 m and various beam spacings: 1.0 m, 1.8 m, 3.6 m, and 7.2 m. The findings of this research are consistent with those of Halawa and Safwat [23], who investigated smoke propagation under a beam height of 1 m and beam spacings of 5 m, 10 m, 15 m, and 20 m. Their results indicated that with a 5 m beam spacing the distance of smoke spread decreased by 79.4% compared to scenarios without beams. Additionally, when compared to larger beam spacings, there was a higher concentration of smoke, the high-temperature region encroached upon the personnel activity area, and the minimum height above the ground was only 1.8 m.

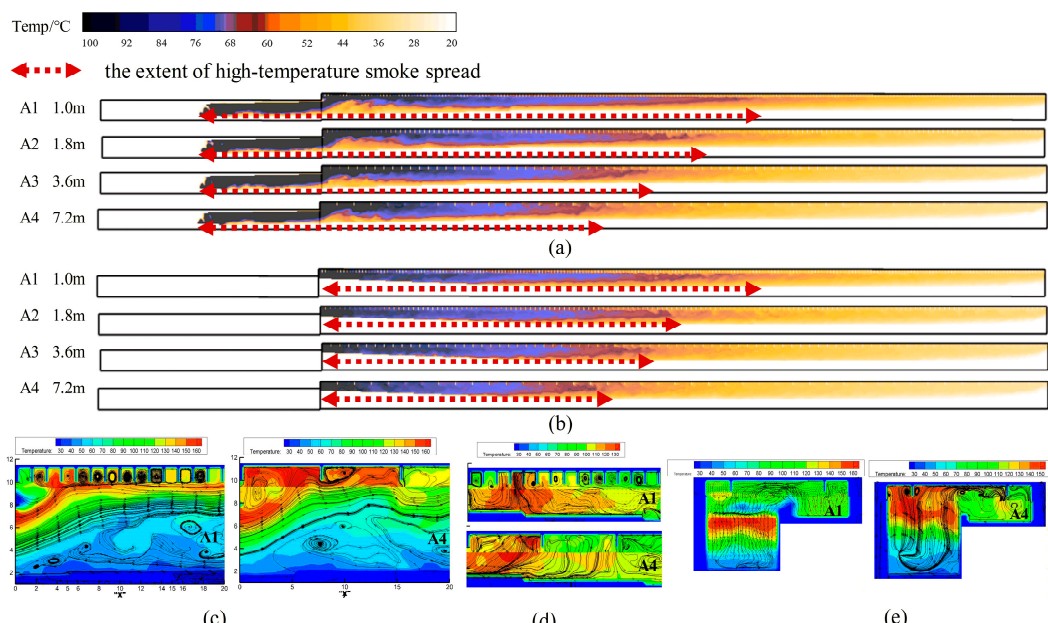

**Figure 8.** Temperature distribution contour (A means fire source located in the immersed tunnel): (**a**) Temperature distribution cloud at Y = −3.75 m slice location; (**b**) Temperature distribution cloud at Y = 4.25 m slice location; (**c**) Temperature-flow velocity distribution cloud at Y = −3.75 m slice location (Convergence Section Tunnel); (**d**) Temperature-flow velocity distribution cloud at Y = 4.25 m slice location (Entrance to the highway service road); (**e**) Temperature-flow velocity distribution cloud at X = 1.3 m slice location (Tunnel variable cross section).

As shown in the comparison of the heights of the high-temperature region above the four-lane section in Figure 8a,b, smaller girder lattice spacing of 1.0 m and 1.8 m resulted in a thinner smoke layer due to a lower kinetic energy and less interference from dense beams. In contrast, larger beam spacing 3.6 m and 7.2 m allowed smoke to move more freely, accumulating kinetic energy. This led to more significant disruptions to the stability of the smoke layer, resulting in an increased smoke layer thickness. Figure 8c–e demonstrated that smaller lattice spacing of 1.0 m generated a small vortex within the beam lattice, ensuring a stable smoke layer. However, with larger spacing of 7.8 m, flow lines fluctuated considerably within 20 m of the converging section entrance, primarily due to the tunnel's low height and the absence of longitudinal ventilation. Consequently, smoke filled most of the tunnel space.

In summary, these observations underscored the substantial impact of beam lattice spacing on smoke behavior, providing valuable insights for controlling the temperature and smoke distribution in complex tunnels.

As shown in Figure 9a, when the fire source was in the confluence section, the temperature at a 2 m height upstream of the fire source closely matched the ambient temperature. The longitudinal wind speed of 2 m/s effectively controlled the smoke spread upstream of the fire source. Downstream of the fire source, lower beam grid spacing resulted in a decrease in the longitudinal temperature distribution. Smaller beam grids had less impact on the smoke spread disturbance, and the 2 m height temperature was less affected by high-temperature smoke.

As smoke spread downstream from the fire source, heat was lost as smoke circulated and drew in cold air. This gradual temperature decrease continued with a greater distance from the fire source. The highway tunnel and confluence section did not have wall panels separating the top space, allowing smoke to spread laterally above the 17 m wide highway tunnel. In the highway tunnel, where mechanical ventilation was absent, the maximum temperature decreased significantly. Figure 9b shows temperature drops along both sides of the highway tunnel. Due to the influence of longitudinal ventilation, the smoke not only

spreads laterally to the road auxiliary section but also extends longitudinally along the tunnel downstream of the fire source. Consequently, the highest temperature above the road auxiliary section does not occur at the same longitudinal position as the fire source but shifts downstream.

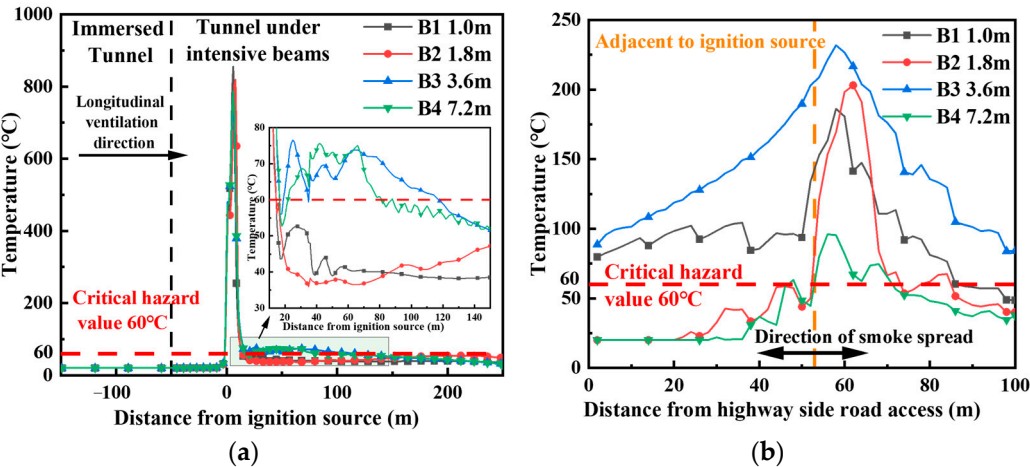

(**a**)　　　　　　　　　　　　　　(**b**)

**Figure 9.** Longitudinal temperature distribution at 2 m height of tunnel (fire source located in the confluence section): (**a**) tunnel mainline section, the green area shows the details of the change; (**b**) auxiliary road section.

As shown in Figure 10a,b, a longitudinal ventilation wind speed of 2 m/s effectively prevented smoke from reaching the concealed buried section. When the beam grid spacing was 3.6 m, the high-temperature region had the shortest spreading distance but increased in thickness due to the greater perturbation of the flue gas movement.

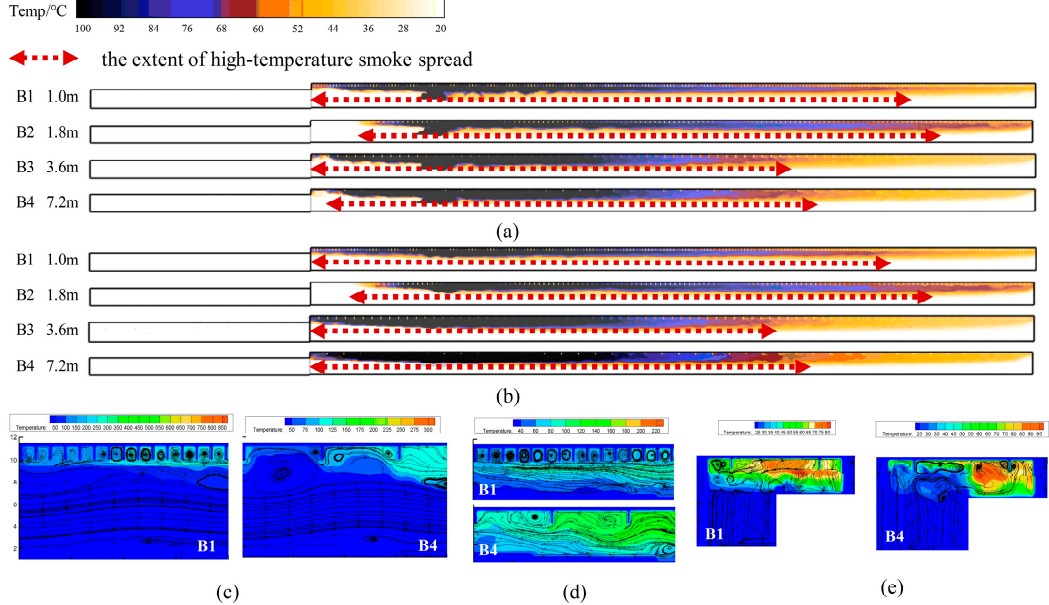

**Figure 10.** Temperature distribution contour (B means fire source located in the confluence section): (**a**) Temperature distribution cloud at Y = −3.75 m slice location; (**b**) Temperature distribution cloud at Y = 4.25 m slice location; (**c**) Temperature-flow velocity distribution cloud at Y = −3.75 m slice location (Convergence Section Tunnel); (**d**) Temperature-flow velocity distribution cloud at Y = 4.25 m slice location (Entrance to the highway service road); (**e**) Temperature-flow velocity distribution cloud at X = 1.3 m slice location (Tunnel variable cross section).

In Figure 10c,d, the flue gas return length was significantly shorter, with a 7.2 m beam spacing compared to a 1.0 m beam spacing, resulting in a thicker flue gas layer. In the highway auxiliary roadway, high-temperature smoke filled the entire tunnel with a 7.8 m beam grid spacing, while a 1.0 m spacing led to the formation of small smoke vortices between the beams.

Figure 10e shows a consistent high-temperature smoke layer thickness between the highway subway and the confluence section with a 1.0 m girder grid spacing. However, a 7.8 m girder grid spacing reduced the smoke return velocity in the confluence section compared to the auxiliary road section, partly due to the absence of smoke evacuation facilities in the latter. This emphasized the significant impact of larger girder grid spacing on obstructing smoke flow.

When the fire source was located on the auxiliary road, Figure 11a indicated that temperatures were higher in the confluence section near the fire source. As smoke spread to the four lanes, temperatures gradually decreased. Interestingly, larger beam spacings resulted in higher temperatures than smaller spacings. This trend was consistent with the scenario where the fire source was located in the confluence section.

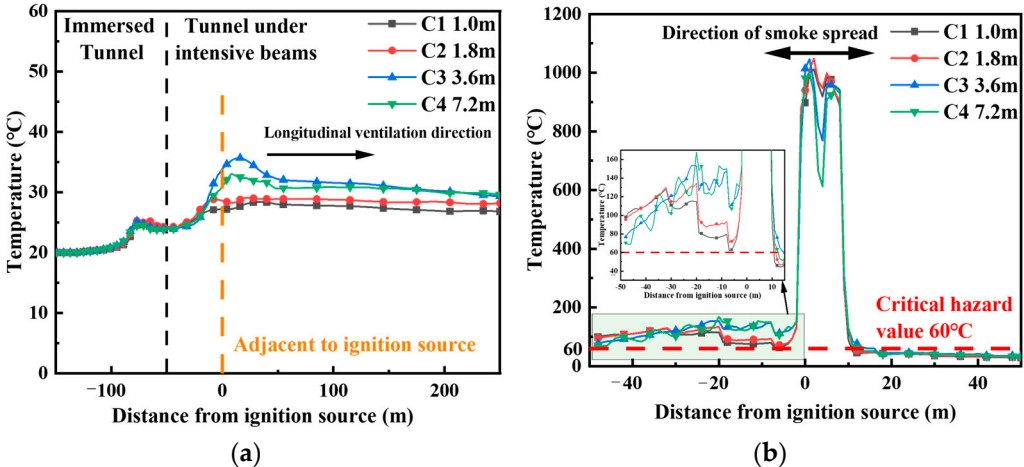

**Figure 11.** Longitudinal temperature distribution at 2 m height of tunnel (fire source located in the auxiliary road): (**a**) tunnel mainline section; (**b**) auxiliary road section, the green area shows the details of the change.

Comparatively, when the fire source was in the highway auxiliary road it had less influence on the temperature field of the adjacent lane. The smaller tunnel height in the highway section caused higher temperatures at 2 m after the smoke settled in the same longitudinal position as the fire source. Additionally, a 2 m/s longitudinal ventilation wind speed in the confluence section cooled high-temperature smoke while suppressing its spread. This lowered the temperature in the confluence section, introducing variability in how the fire source's location affected the longitudinal temperature distribution of neighboring road sections.

In Figure 11b, with the fire source in the highway auxiliary road, natural smoke spread occurred. Smaller beam spacings resulted in less perturbation of the smoke layer and greater stability. Smoke accumulated in the upper region of the tunnel, and the upstream temperature of the fire source was lower when compared to scenarios with larger beam spacings. Downstream of the fire source, the longitudinal temperature distribution at the 2 m height remained close to the ambient temperature.

In Figure 12a,b, it was observed that the high-temperature region above the four lanes shortened as the spacing of the beam lattice increased. Notably, the high-temperature region was shortest when the beam lattice spacing was 3.6 m, allowing for the most effective control of smoke spread. When the beam lattice spacing exceeded 1.8 m, it became apparent that the lattice disturbed the smoke, resulting in a thicker smoke layer. Figure 12c,d demonstrated

that larger beam lattice spacing led to a wider region of flow velocity fluctuation and a thicker smoke layer.

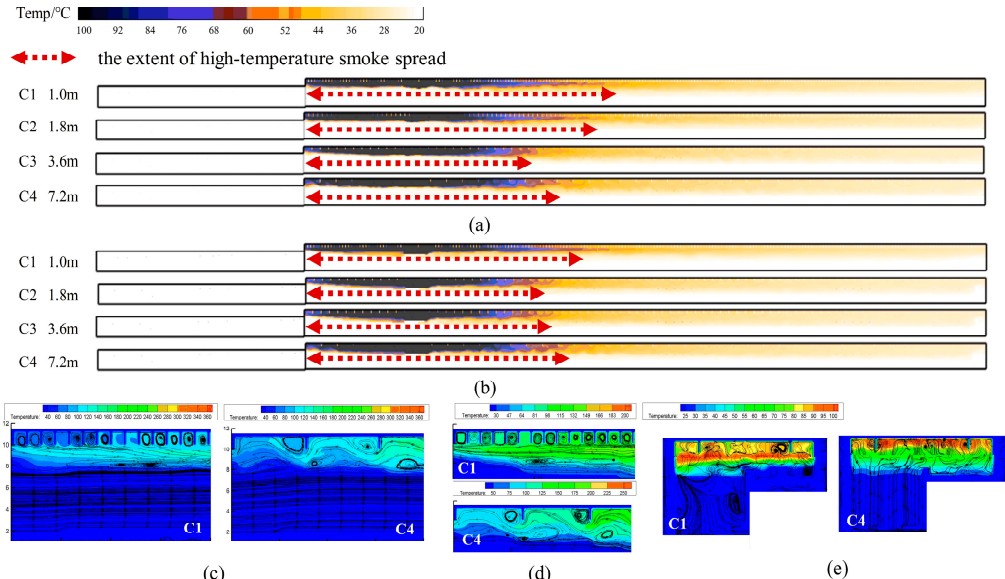

**Figure 12.** Temperature distribution contour (C means fire source located in the auxiliary road): (**a**) Temperature distribution cloud at Y = −3.75 m slice location; (**b**) Temperature distribution cloud at Y = 4.25 m slice location; (**c**) Temperature-flow velocity distribution cloud at Y = −3.75 m slice location (Convergence Section Tunnel); (**d**) Temperature-flow velocity distribution cloud at Y = 4.25 m slice location (Entrance to the highway service road); (**e**) Temperature-flow velocity distribution cloud at X = 1.3 m slice location (Tunnel variable cross section).

When the fire source was positioned within the four-lane section, and to better comprehend the smoke dispersion pattern into the buried section, confluence section, and highway auxiliary road section, the fire source was placed at the center of the tunnel, 10 m from the exits of both the confluence section and the highway auxiliary road section.

As illustrated in Figure 13a, under a longitudinal wind speed of 2 m/s, the temperature in the buried section remained close to ambient. However, within a range of 25 m upstream of the fire source, temperatures at a 2 m height exhibited an increase. With the expansion of the beam grid spacing, the smoke experienced greater disturbance, disrupting the smoke layer structure and elevating tunnel temperatures.

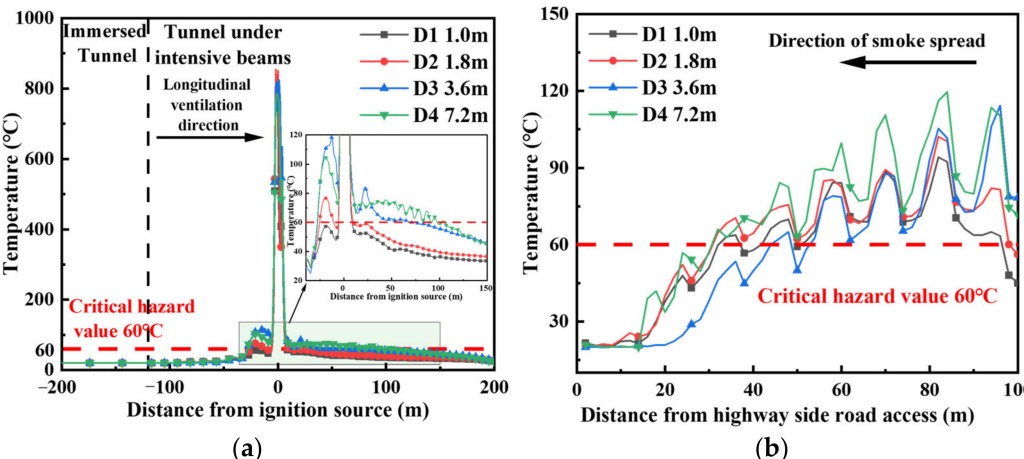

**Figure 13.** Longitudinal temperature distribution at 2 m height of tunnel (fire source located in the four-lane section of the tunnel): (**a**) tunnel mainline section, the green area shows the details of the change; (**b**) auxiliary road section.

Figure 13b revealed significant fluctuations in the longitudinal temperature distribution at a 2 m height within the highway access road. Smoke spreading from the four lanes to the highway access road resulted in disordered smoke layer structures due to the absence of uniform longitudinal ventilation. There was no clear stratification, and the disturbance and fluctuation of smoke were more pronounced with larger girder grid spacing.

Figure 14a,b illustrates that when the fire source was situated within the four-lane section, smoke was contained within the confluence section, gathering at the tunnel's upper region. The length of the high-temperature zone in the four-lane area shortened and then increased with the growing lattice spacing, with the shortest high-temperature size observed at a spacing of 3.6 m. As with other fire source locations, the smoke layer thickness within the tunnel significantly increased with the expanding lattice spacing.

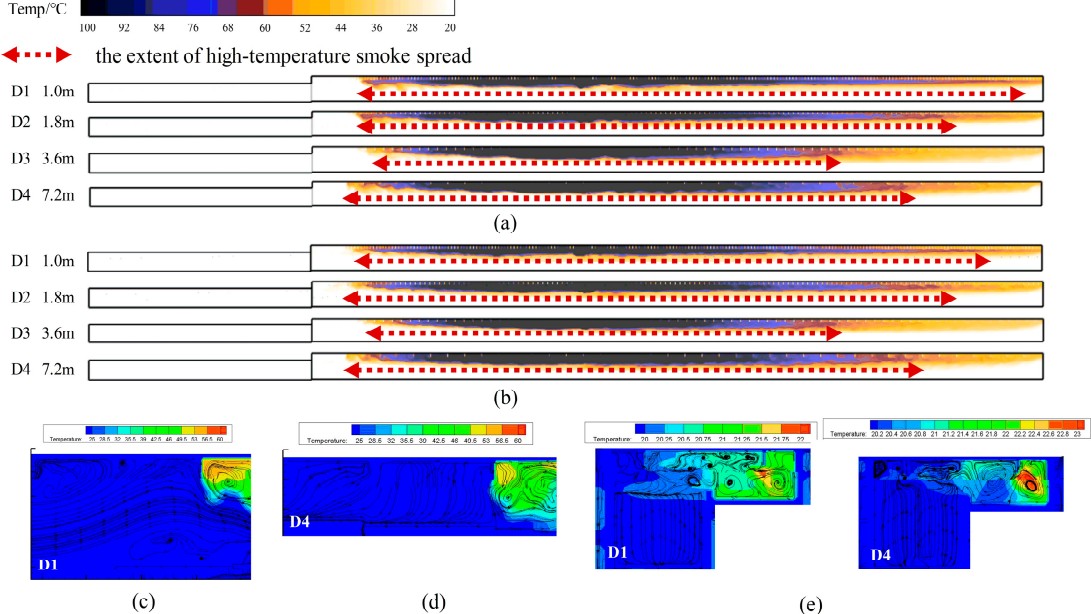

**Figure 14.** Temperature distribution contour (D means fire source located in the four-lane section of tunnel): (**a**) Temperature distribution cloud at Y = −3.75 m slice location; (**b**) Temperature distribution cloud at Y = 4.25 m slice location; (**c**) Temperature-flow velocity distribution cloud at Y = −3.75 m slice location (Convergence Section Tunnel); (**d**) Temperature-flow velocity distribution cloud at Y = 4.25 m slice location (Entrance to the highway service road); (**e**) Temperature-flow velocity distribution cloud at X = 1.3 m slice location (Tunnel variable cross section).

For a lattice spacing of 1 m, the ambient temperature was essentially maintained in the confluence section and within a 20 m range at the entrance of the highway auxiliary road. In contrast, when the lattice spacing was 7.2 m only the temperature and flow rate distribution cloud diagram was provided (Figure 14c,d). This cloud diagram revealed substantial disruption in the smoke layer, mainly when the beam lattice spacing was 7.8 m.

With a 1.0 m beam lattice spacing, the smoke exhibited a faster reflux rate within the highway auxiliary road than in the confluence section, as shown in Figure 14e. When the beam grid spacing was 7.2 m, the smoke predominantly spread against the side wall of the highway aisle.

Figure 15, focusing on the fire source located in the confluence section, provided insights into smoke movement behavior. The smoke spread horizontally to the highway auxiliary road section and longitudinally along the tunnel ceiling.

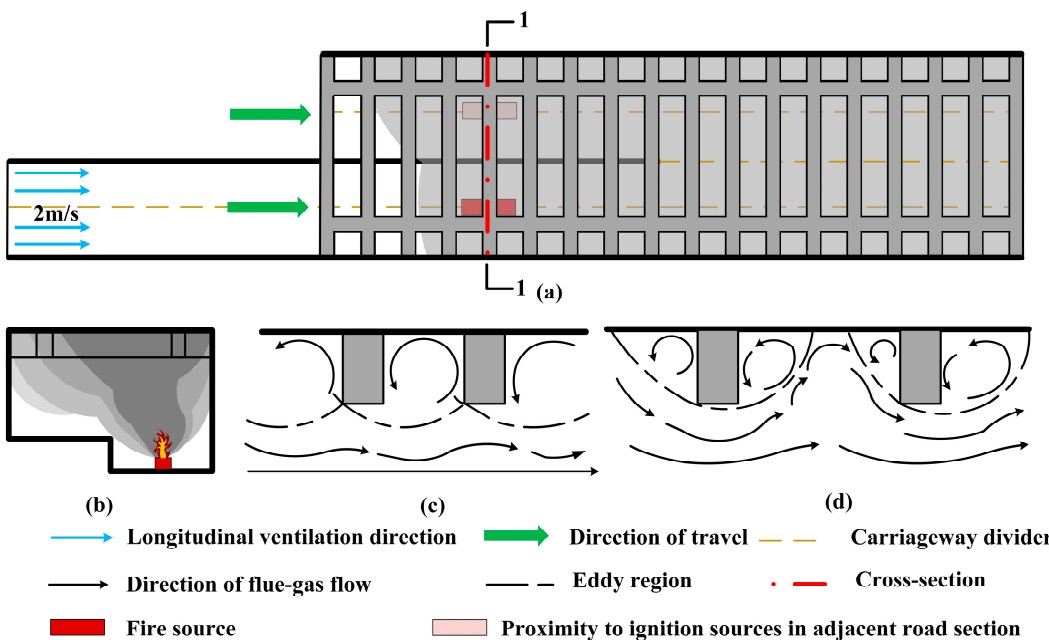

**Figure 15.** Schematic diagram of smoke spread and evacuation direction: (**a**) Top view of smoke spread; (**b**) 1-1 section view; (**c**) Skimming flow (Beam spacing 1 m, 1.8 m); (**d**) Isolated roughness flow (Beam spacing 3.6 m, 7.2 m).

A critical ventilation rate of 2 m/s at the entrance of the immersed tube section ensured a smoke-free condition in the immersed tube section for all fire source locations except when the fire occurred in the immersed tube section itself. Due to the uneven wind speed distribution beneath the girder, the smoke front might exhibit a bias towards the highway auxiliary road. The identification of fire source locations in adjacent sections is essential, especially in areas where highway side roads and confluence sections are closely situated. This labeling serves to distinguish and compare variations in the fire temperature field and the pattern of smoke spread within neighboring sections.

Smoke propagation in the tunnel section under the beam was influenced by beam grid spacing, resulting in two distinct smoke movement patterns, as shown in Figure 15c,d. Smaller beam grid spacing led to the formation of smoke vortexes between the beam grids, which accumulated less momentum and had a more negligible impact on the stability of the smoke layer. This resulted in lower temperatures at the 2 m height, providing a relatively safe evacuation environment.

In contrast, larger beam spacing encouraged extensive smoke development between the beams. This significantly disrupted the stability of the smoke layer, leading to an increased smoke and air volume suction, more significant smoke generation, and a thicker smoke layer. Consequently, the temperature at the 2 m height rose, creating a more hazardous evacuation environment.

It is important to note that the longitudinal temperature distribution at the 2 m height of the highway auxiliary road was influenced by various factors, including cross-section changes, the low tunnel height, and the uneven ventilation airflow. The impact of beam lattice spacing on this distribution varied.

### 3.2. Distribution of Tunnel Visibility at Different Fire Locations

Figure 16 highlighted that when the fire source was in the immersed tube section visibility was generally better with wider beam spacing, as indicated by the green and blue lines. This phenomenon resulted from several factors. When the fire source was in the immersed tube section, the smoke temperature was lower as it spread into the converging area and the highway service road, resulting in weaker smoke thermal buoyancy. The substantial beam grid spacing acted as a barrier that trapped most of the smoke between

the beam grids. This containment reduced the settled smoke amount, ultimately enhancing visibility at a height of 2 m.

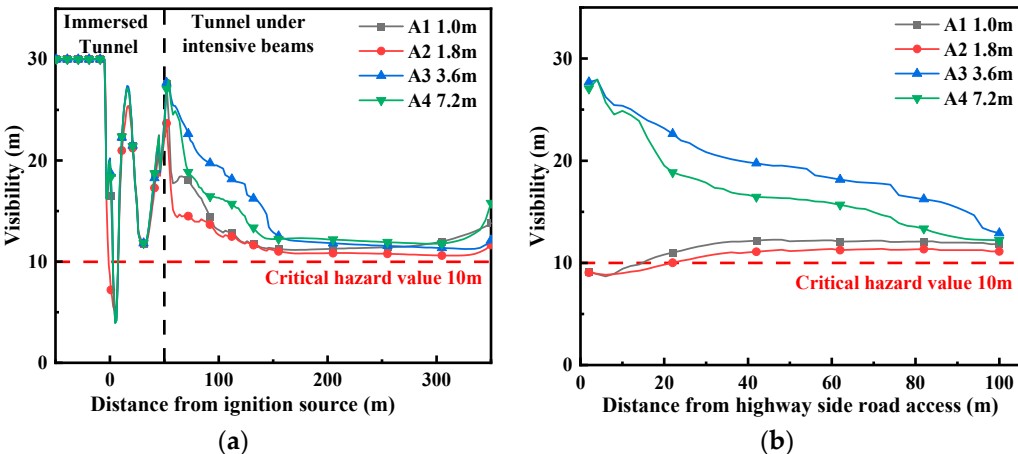

(a)

(b)

**Figure 16.** Visibility distribution at 2 m height in the tunnel (fire source located in the immersed tunnel): (**a**) tunnel mainline section; (**b**) auxiliary road section.

In scenarios where the fire source was located in the four-lane section, the smoke's temperature in the tunnel section beneath the beams was high. Through Figure 17, it can be observed that when the beam grid spacing was larger the visibility downstream of the fire source was reduced. This could be attributed to two main factors. Smoke lost heat during longitudinal spreading, leading to a weakened thermal buoyancy, increased smoke settling, and reduced visibility at a height of 2 m. Larger beam spacing had a stronger blocking effect on smoke, disrupting the stable smoke layer structure and further reducing visibility. It is worth noting that, except for the case where the fire source was on the highway side road, visibility upstream of the fire source remained high and was largely unaffected by the smoke.

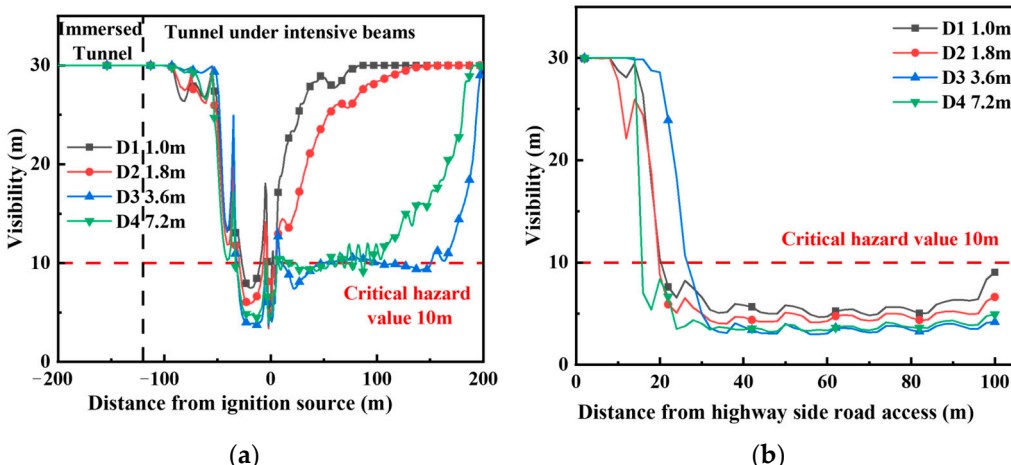

(a)

(b)

**Figure 17.** Visibility distribution at 2 m height in the tunnel (fire source located in the four-lane section of tunnel): (**a**) tunnel mainline section; (**b**) auxiliary road section.

When the fire source was situated in the immersed tube section, the impact of beam lattice spacing on smoke behavior became evident. In Figure 18a, the rate of the smoke spread decelerated as the beam lattice spacing widened, emphasizing the enhanced smoke-blocking effect of the beam lattice.

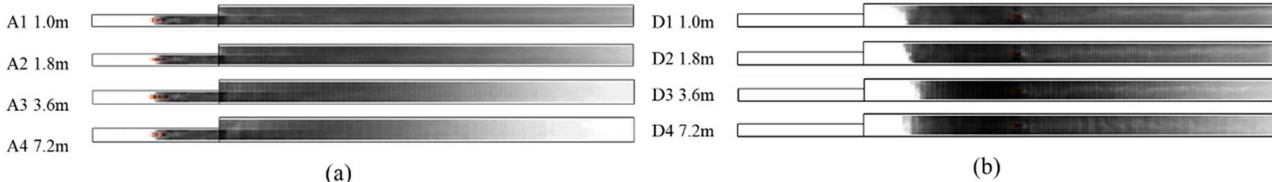

**Figure 18.** The top view of smoke spread at 400 s: (**a**) A represents the fire source in the immersed tunnel; (**b**) D represents the fire source in the four-lane section of the tunnel.

Examining the temperature distribution cloud in Figure 18b, it was observed that when the fire source was positioned in the four-lane section the rate of smoke return upstream of the fire source diminished with the increasing beam lattice spacing, reaching the slowest rate when the spacing was 3.6 m. As the beam lattice spacing increased, the smoke return rate in the highway auxiliary road also decreased. Furthermore, the smoke return length in the highway side road surpassed that of the confluence section, with the smoke front tilting towards the side of the highway side road. This was primarily due to the stronger effect of longitudinal ventilation in the confluence section, which helped to suppress the smoke backflow from this region.

### 3.3. Tunnel Fire Safety Analysis

In the absence of rafters above the tunnel, the temperature at a 2 m height upstream of the fire source remained at the ambient temperature, except when the fire source was positioned at the roadside channel, as depicted in Figure 19a. Figure 19b illustrates that when the fire source was located in the highway section, high-temperature smoke spilled out from the highway section exit due to the influence of the slope and variable cross-section. This caused the temperature upstream of the fire source to exceed the ambient temperature, with the maximum temperature above the fire source reaching 1000 °C. When the fire source was situated in the converging and highway sections, the temperature at a 2 m height within the tunnel remained at the ambient temperature. When the fire source was positioned in the converging and highway sections, its impact on the temperature distribution in neighboring tunnels was minimal, ensuring safe evacuation conditions.

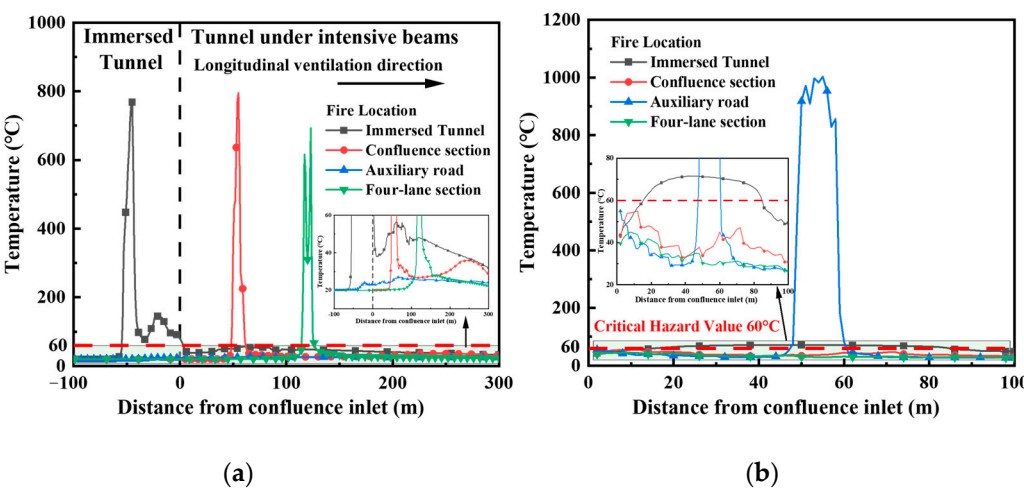

**Figure 19.** Longitudinal temperature distribution at 2 m in the tunnel at different fire locations, the green area shows the details of the change: (**a**) tunnel mainline section (**b**) auxiliary road section.

When the fire source was situated in the immersed tube section, the smoke in this section was influenced by longitudinal ventilation and spread into the tunnel section under the cover. The confluence and road auxiliary sections experienced significant smoke accumulation with relatively high temperatures, presenting the most dangerous conditions. These sections had relatively low visibility inside the tunnel, as indicated by the black lines

in Figure 20. However, when the fire source was located in other positions, the visibility at a 2 m height was mostly unaffected in the remaining sections, except for reduced visibility near the fire source.

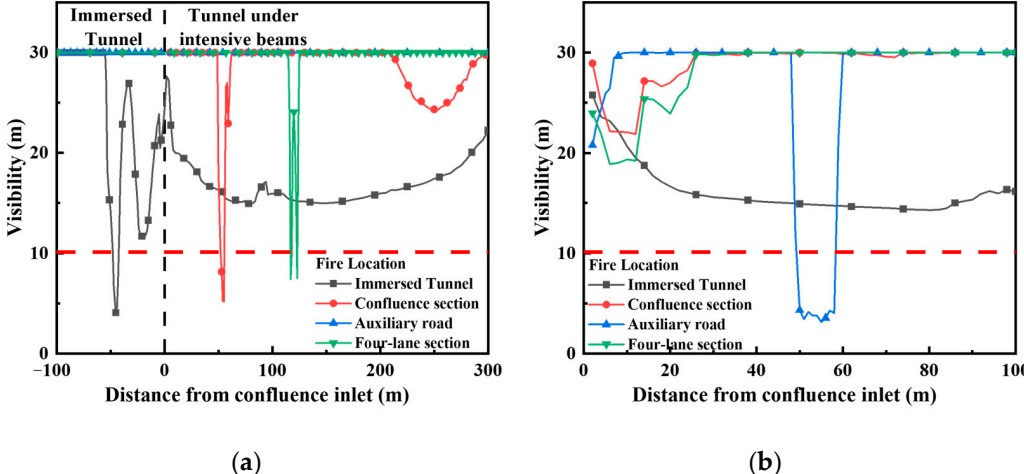

**Figure 20.** Visibility distribution at 2 m in the tunnel at different fire locations, the green area shows the details of the change: (**a**) tunnel mainline section; (**b**) auxiliary road section.

In complex tunnels with upper covers, the degree of danger varied depending on the location of the fire within the tunnel. Additionally, different beam spacing at the top affected the behavior of the fire smoke spread. In this study, the fire hazard range within the tunnel was defined based on the *SFPE Fire Engineering Handbook* [25], as shown in Table 3. A critical hazard value was set, considering a temperature greater than 60 °C or visibility less than 10 m. As the range where the temperature reached the critical hazard value was larger than that of the visibility, the critical temperature value was used to determine hazardous areas in the evacuation environment.

**Table 3.** Tunnel evacuation hazardous area statistics.

| No. | Fire Location | Beam Spacing/m | Hazard Range/m | | | | |
|---|---|---|---|---|---|---|---|
| | | | Immersed Tunnel | Confluence Section | Auxiliary Road | Four-Lane Section | Sum |
| A0 | | None | 55 | SAFE | 70 | SAFE | 125 |
| A1 | | 1 | 58 | 53 | SAFE | 25 | 136 |
| A2 | Immersed tunnel | 1.8 | 58 | 69 | SAFE | 25 | 152 |
| A3 | | 3.6 | 58 | 83 | 104 | 20 | 265 |
| A4 | | 7.2 | 58 | 85 | 16 | SAFE | 159 |
| B0 | | None | SAFE | 14 | SAFE | SAFE | 14 |
| B1 | | 1 | SAFE | 16 | 24 | SAFE | 40 |
| B2 | Confluence section | 1.8 | SAFE | 16 | 104 | SAFE | 120 |
| B3 | | 3.6 | SAFE | 49 | 34 | 68 | 151 |
| B4 | | 7.2 | SAFE | 46 | 88 | 38 | 172 |
| C0 | | None | SAFE | SAFE | 12 | SAFE | 12 |
| C1 | | 1 | SAFE | SAFE | 61 | SAFE | 61 |
| C2 | Auxiliary road | 1.8 | SAFE | SAFE | 62 | SAFE | 62 |
| C3 | | 3.6 | SAFE | SAFE | 66 | SAFE | 66 |
| C4 | | 7.2 | SAFE | SAFE | 63 | SAFE | 63 |
| D0 | | None | SAFE | SAFE | SAFE | 13 | 13 |
| D1 | | 1 | SAFE | SAFE | 66 | 15 | 81 |
| D2 | Four-lane section of tunnel | 1.8 | SAFE | 10 | 70 | 26 | 106 |
| D3 | | 3.6 | SAFE | 13 | 62 | 91 | 166 |
| D4 | | 7.2 | SAFE | 13 | 74 | 113 | 200 |

Table 3 summarizes the impact of fire within the tunnel, considering the presence of top beam lattice structures. The most hazardous situation occurred when the fire source was in the immersed tube section with a beam lattice spacing of 3.6 m, resulting in a hazardous area extending up to 265 m. This was the most dangerous condition among all the scenarios considered. In contrast, when the fire source was in the road service section its impact on the fire safety of other sections was relatively lower. However, the road service section itself had a larger hazardous area, requiring a focus on rescue measures.

When the fire source is in the immersed tube section, vehicles downstream of the fire source in the main tunnel can still be driven directly to the tunnel exit, ensuring personnel safety. Although the tunnel temperature exceeds the critical danger value for over 200 m, the faster evacuation speed of vehicles is a safety assurance. In the four-lane section, where visibility is limited, careful road condition management during evacuation is essential to prevent traffic accidents blocking the escape route. A longitudinal wind speed of 2 m/s effectively prevents smoke from flowing upstream of the fire, ensuring a safe evacuation.

The tunnel temperature and visibility remain within safe ranges within the auxiliary road section and with a beam grid spacing of less than 3.6 m. In this scenario, it is advisable to contain the fire's impact within the auxiliary road, allowing vehicles in that section to be driven directly out of the tunnel.

In typical tunnels, a fundamental evacuation principle dictates that vehicles cannot pass the location of the fire source during an emergency. Therefore, occupants upstream of the fire source must exit their cars to escape, while those downstream can drive to safety. However, in the case of the complex tunnel discussed in this paper, with various lanes divided into converging and highway auxiliary sections, distinct road conditions exist in different areas. This necessitates a redefinition of the upstream and downstream safety zones concerning the fire source.

Figure 21 illustrates these safety zones and vehicle access points when the fire source is in the converging section. In this case, when the fire source is in the confluence section, the temperature and visibility upstream of the fire source remain within safe limits. In such situations, individuals should promptly exit their vehicles and evacuate through designated doors or exits. In cases where the beam spacing is less than 3.6 m, the evacuation environment downstream of the fire source is relatively safe. Vehicles can continue to exit the tunnel in the direction of travel, and individuals can also use the pedestrian passages on both sides of the tunnel in emergencies, such as vehicle breakdowns.

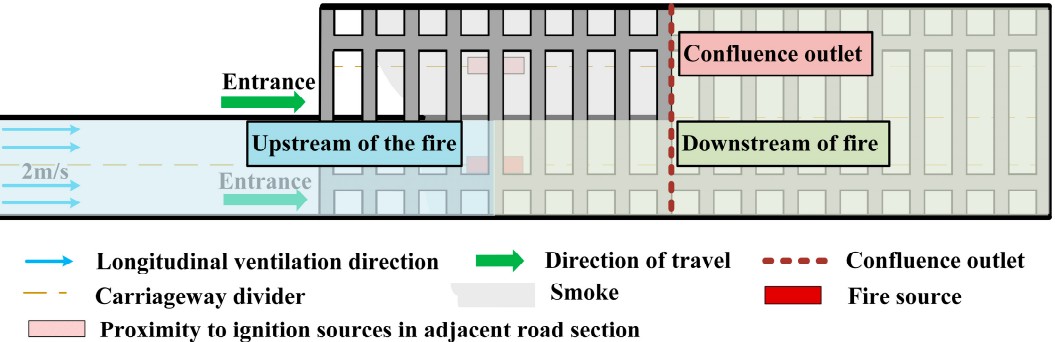

**Figure 21.** Schematic diagram of the relative positions of people in the direction of evacuation (fire source located in the confluence section).

For scenarios with a larger gap between the beams, the evacuation environment inside the tunnel becomes more perilous. In these instances, a lane should be designated for specific rescue purposes if a vehicle experiences mechanical failure and requires assistance evacuating occupants. In the highway auxiliary road section, the temperature is high, and smoke spreads from the position of the adjacent fire source to both sides. However, this section is not obstructed by the fire source, enabling vehicles to continue driving past the fire source. Individuals should aim to avoid abandoning their cars for escape in the

challenging conditions of a high temperature and darkness caused by the fire. Instead, they should heed the commands of fire and rescue personnel to drive their vehicles out of the high-temperature zone quickly.

When the fire source is in the highway auxiliary road, the temperature and visibility in the immersed tube section, confluence section, and mainline of the tunnel are within safe levels. Vehicles in the mainline can exit the tunnel directly. Downstream cars in the highway auxiliary road can also evacuate directly. However, people and vehicles upstream of the fire source are exposed to a more dangerous fire environment, as indicated in Figure 22.

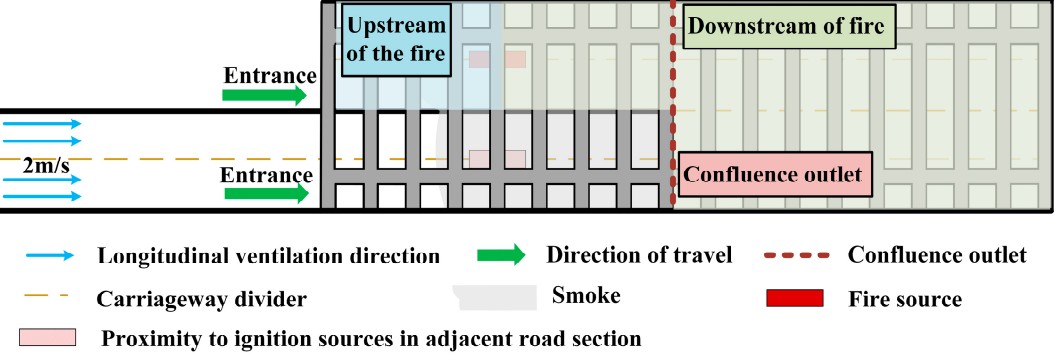

**Figure 22.** Schematic diagram of the relative positions of people in the direction of evacuation (fire source located in the auxiliary road).

In such cases, firefighters must act promptly to cool the area upstream of the fire source, conduct swift search and rescue operations, and provide gas masks and fire blankets to assist in the evacuation.

If the fire source is in the four-lane section, the temperature and visibility in the immersed tube and converging sections are within safe limits. Nearby individuals can evacuate through designated points. Vehicles downstream of the fire source in the four-lane section should exit the tunnel quickly.

While the entrance section of the road has a slightly lower visibility compared to the rest of the road section, it still exceeds the critical risk value of 10 m. At the exit section, the temperature is relatively high but remains below the critical risk value of 60 °C. It is advisable for vehicles to use the right lane during evacuations. In firefighting and rescue operations, attention should be given to measures on the roadside, including deploying exhaust fire trucks to control smoke, increase visibility, reduce the temperature, and assist vehicles and personnel in their escape.

## 4. Conclusions

In conclusion, the presence of closely spaced beams on the tunnel's ceiling has a notable impact on the tunnel's fire hazard. After considering various fire source locations and beam grid spacing and analyzing fire risks based on the temperature and visibility, the following conclusions can be drawn:

1.  Increased beam grid spacing enhances the smoke storage capacity and reduces the distance smoke can spread but disrupts the stability of the smoke layer, resulting in a thicker smoke layer, higher temperatures, and reduced visibility at a 2 m height. Increased spacing between the beams undermines the stability of the smoke layer, causing the deposition of high-temperature smoke, a rise in the temperature of the fire scene, and diminished visibility. This heightened risk poses challenges for the evacuation environment during tunnel fires.

2.  Regardless of the location of the fire, larger beam grid spacing results in a slower rate of smoke spread. A spacing of 3.6 m minimizes the range of the smoke spread. However, when the fire source is in the four-lane section, due to uneven ventilation

wind speed distribution, the smoke return rate above the highway auxiliary road is faster than in the converging section.

3. The beam lattice structure on the tunnel's ceiling heightens the fire hazard, especially with larger beam lattice spacing, which signifies a more perilous fire evacuation environment. In such scenarios, ensuring the safe evacuation of vehicles and individuals becomes crucial, necessitating additional firefighting assistance. This is particularly critical when the fire originates in the four-lane section. With a girder spacing of 7.2 m, the evacuation hazardous area has expanded by 197 m, approximately 46% of the tunnel length, compared to the condition without beam grids. Furthermore, the evacuation environment in the auxiliary roadway becomes perilous, demanding heightened attention.

The central focus of this paper is ensuring the safe evacuation of individuals, although the exploration and analysis of tunnel fire phenomena are relatively limited. This study endeavors to deepen our understanding of the effects of various fire scenarios within intricate tunnel structures, particularly on the distribution of the evacuation temperature, smoke, and visibility in neighboring tunnels. It also delivers essential insights into how closely spaced beams impact personnel safety during evacuations at the tunnel ceiling. Through meticulous investigation, this study aims to provide scientific guidance and recommendations for relevant emergency rescue strategies and the design of tunnel structures. In future research endeavors, we aim to delve deeper into understanding how the spacing of dense beams at the tunnel's apex and the beam height impact factors such as the longitudinal temperature distribution, smoke back-layering length, critical velocity, and density leap beneath the tunnel ceiling during a fire. Our goal is to conduct a quantitative analysis of fire hazards within the tunnel structure. This approach will provide scientifically sound and easily comprehensible guidance for both engineering design and emergency rescue management in related scenarios.

**Author Contributions:** Conceptualization, L.W. and Z.X.; methodology, Z.Y.; software, S.C.; validation, S.C., Y.X., and S.F.; investigation, J.Z.; resources, L.W. and H.T.; data curation, S.C. and S.F.; writing—original draft preparation, L.W.; writing—review and editing, Z.Y. and Z.X.; visualization, S.F.; supervision, Z.X. and Z.Y.; project administration, L.W.; funding acquisition, H.T. and Y.X. All authors have read and agreed to the published version of the manuscript.

**Funding:** This research was funded by Guangzhou Municipal Engineering Design & Research Institute Co.Ltd (GMEDRI), grant number 2021-092-FK.

**Institutional Review Board Statement:** Not applicable.

**Informed Consent Statement:** Not applicable.

**Data Availability Statement:** Data are contained within the article.

**Acknowledgments:** The authors acknowledge the support of Central South University's High-Performance Computing Center.

**Conflicts of Interest:** The authors declare no conflict of interest.

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
