# Peer review of "Fire Hazard Analysis on Different Fire Source Locations in Multi-Segment Converging Tunnel with Structural Beams"

_fire, doi:10.3390/fire6110444_

Round 1

Reviewer 1 Report

Comments and Suggestions for Authors

The author presents a numerical simulation study on tunnel fire behavior with beam structures and multiple-section convergence, considering various fire source locations. This study is insightful, and my comments are shown below:

1.     The use of tunnels with overhead beam structures is relatively uncommon. It would be beneficial to provide some images of similar completed tunnels.

2.     Some of the temperature distribution figures in Section 3.1 are not clear.

3.     The linkage between Figures 15 and 16 in Section 3.2 is somewhat weak, and the accompanying analysis is somewhat verbose. I suggest revising the analysis to highlight the primary research findings.

4.     In Figures 14, 19, and 20, virtual fire sources are depicted without corresponding explanations in the text. It remains unclear how these virtual fire sources were defined and their practical significance should be emphasized.

5.     The definition of upstream and downstream regions of the fire source in Figure 20 is not correct. Please check this.

6.     This study indicates that large beam space reduces smoke backflow length, which positively impacts smoke control in fires. However, the conclusion indicates that smoke height would be smaller for larger beam distance which means high risk for evacuating.

Comments on the Quality of English Language

Minor editing of English language is required.

Reviewer 2 Report

Comments and Suggestions for Authors

This study examined potential fire hazards within complex tunnels characterized by various cross-sections, sloped structures, and closely spaced upper cover beams. Four specific fire source locations within the tunnel were investigated with useful conclusions obtained, comments can be found below,

1.      The necessity for conducting this research should be clearly presented. There is no logic in the Introduction part.

2.      The motivation of the paper and the literature review should be strengthened. The authors are asked to conduct a deeper literature review for the background of the research.

3.      Which turbulence model have you used in this study? Please be specific.

4.      CFD simulation accuracy is easily influenced by numerical setting, thus numerical settings should be clearly described.

5.      Why do you present temperature distribution contour of Y ranging from -4 to 4.5 m in Fig. 7? Please give both the smoke temperature and the contour image for each section.

6.      In Fig. 8, the location of fire source is not coincide with the location with maximum temperature, please give the reasons.

Comments on the Quality of English Language

Minor editing of English language required

Reviewer 3 Report

Comments and Suggestions for Authors

Dear Authors,

In line with the proofreading criteria of the publisher, I prepared a reviewer’s report, which would be as follows:

The content of the proposed paper on high level meets the objectives of the journal and special issue.

Using the scientific methods (Fire Dinamics Simulation) applied in accordance with the author’s scientific objectives resulted useful scientific achievements.

The main strength of the study is that the author’s scientific work may have positive impact on optimizing tunnel fire safety knowledge and strategy. The present work of the authors provides useful assistance to both fire protection authorities and business organizations.

The references used in the main chapters are relevant and assist the reader to understand the authors proposals. The illustrations used are regular and correct.

In addition to acknowledging the high-quality work, I propose the following amendments:

-          In my opinion, the type of the paper is Article.

-          At the end of Section “1. Introduction”, the main objectives of this article should be clearly and in detail presented.

-         It can be considered to rename Section 3 into “Results and Discussion”.

-          4. Conclusions. It is recommended to specify shortly the possibilities of international theoretical and practical applicability of the present  research, as well as the possible directions of future research projects. 

Based on the above, after the revision of the article, I suggest the publication of reviewed article.

Comments on the Quality of English Language

Minor editing of English language required.

Reviewer 4 Report

Comments and Suggestions for Authors

My comments are following:

1. Abstarct section not well written and authors should re-write it and highlight the aims and improtant findings of this study. 

2. Please highlight the new of this study compared to previous studies.

3. Otained results should compare to literature.

4. Conslusion section should revise with more details and outcome of this study. 

Comments on the Quality of English Language

English correction are required. 
